# ADHI: The African Database of Hydrometric Indices (1950-2018)

Yves Tramblay[1]
Nathalie Rouché[1]
Jean-Emmanuel Paturel[1]
Gil Mahé[1]
Jean-François Boyer[1]
Ernest Amoussou[2]
Ansoumana Bodian[3]
Honoré Dacosta[4]
Hamouda Dakhlaoui[5,6]
Alain Dezetter[1]
Denis Hughes[7]
Lahoucine Hanich[8,9]
Christophe Peugeot[1]
Raphael Tshimanga[10]
Patrick Lachassagne[1]

[1] HydroSciences Montpellier, Univ. Montpellier, CNRS, IRD, Montpellier, France

[2] Département de Géographie et Aménagement du Territoire (DGAT) de l'Université de Parakou (UP), BP 123 Parakou, Bénin

[3] Laboratoire Leïdi "Dynamique des Territoires et Développement", Université Gaston Berger (UGB), BP 234 - Saint Louis, Sénégal

[4] Département de Géographie-FLSH, Université Cheikh Anta Diop de Dakar

[5] LMHE, Ecole Nationale des Ingénieurs de Tunis, University of Tunis El Manar, BP 37, 1002 Tunis le Belvédère, Tunisia

[6] Ecole Nationale d'Architecture et d'Urbanisme, University of Carthage, Rue El Quods, 2026, Sidi Bou Said, Tunisia

[7] Institute for Water Research, Rhodes University, South Africa

[8] L3G Laboratory, Earth Sciences Department, Faculty of Sciences & Techniques, Cadi
Ayyad University, BP 459, 40000 Marrakech, Morocco
[9] Mohammed VI Polytechnic University (UM6P), Centre for Remote Sensing and
Application, Morocco
[10] Congo Basin Water Resources Research Center -CRREBaC, University of Kinshasa,
Kinshasa, Democratic Republic of the Congo

51  Revised manuscript


# **Abstract**


The African continent is probably the one with the lowest density of hydrometric stations
currently measuring river discharge, despite the fact that the number of operating
stations was quite important until the 70s. This new African Database of Hydrometric
Indices (ADHI) provides a wide range of hydrometric indices and hydrological
signatures computed from different sources of data after a quality control. It includes
1466 stations with at least 10 years of daily discharge data over the period 1950-2018.
The average record length is 33 years and for over 100 stations complete records are
available over 50 years. With this new dataset spanning most regions of the African
continent, several hydrometric indices have been computed, representing mean flow
characteristics and extremes (low flows and floods), and are made accessible to the
scientific community. The database will be updated on a regular basis to include more
hydrometric stations and longer time series of river discharge. The ADHI database is
available for download at: https://doi.org/10.23708/LXGXQ9 (Tramblay and Rouché,
96 2020).


# **1. Introduction**


There is a growing need for large-scale streamflow archives (Addor et al., 2020;
Hannah et al., 2011), that are extremely useful to evaluate continental land-surface
simulations (Archfield et al., 2015; Newman et al., 2015; Ghiggi et al., 2019; Do et al.,
2020), remote sensing data products (Beck et al., 2017; Brocca et al., 2019; Forootan et
al., 2019; Satgé et al., 2020), develop operational flood or drought monitoring systems
(Alfieri et al., 2020; Harrigan et al., 2020; Lavers et al., 2019; Thiemig et al., 2011), or
evaluate aquifer outflows and characteristics (Dewandel et al., 2003, 2004). In Africa,
the density of active monitoring networks is lower compared to other continents and
there are challenges in the exchange of hydrometric data across countries (Mahé and
Olivry, 1999; Viglione et al., 2010; Mahe et al., 2013; Stewart, 2015; Dixon et al., 2020).

African countries are largely under-represented in large-scale databases such as the
Global Runoff Data Center (GRDC) or the recent GSIM initiative (Do et al., 2018;
Gudmundsson et al., 2018), and/or the time series are mostly not updated. At the
African scale, there is still a lack of coordination for hydrological data collection and
dissemination, despite the launch in 1975 of the UNESCO Intergovernmental
Hydrological Program (IHP) dedicated to water research, water resources management,
as well as education and capacity building. This initiative enhanced the set up and
management of international rainfall and runoff databases at the regional scale of the
FRIEND programs (Van Lanen et al., 2014), but these are still largely not updated.
There is still not enough partnership between the national hydrological services and in
many countries licensing issues prevent the distribution of the data collected.
The density of monitoring networks in Africa has been declining over time; a serious
concern for hydrologists since data acquisition and experimental data analysis remain
central to understand hydrological processes and their spatio-temporal variability
(Hannah et al., 2011; Roudier et al., 2014; Blume et al., 2016; Beven et al., 2020).
There are several reasons for this decline: the budgetary austerity measures imposed
by the international financial institutions, the lack of permanent funding of national
hydrological services, and the typically low number of well-trained technical staff in
these departments (Bodian et al., 2016, 2020; Hannah et al., 2011). As a result,
hydrological monitoring is now often dependent on research projects that cannot
support long term observations. Studies focusing on regional river discharge variability
are rare at the scale of Africa due to the lack of data. For instance, Conway et al. (2009)
could only present a study on a reduced number of representative regional basins, and
Roudier et al. (2014) compared only published anomaly results in their review of climate
change impacts on the hydrology of West Africa.
Since in many cases, there are strict conditions related to the redistribution of un-
processed data (Do et al., 2018), it is very often not possible to provide the complete
time series of discharge data. Nevertheless, hydrological indices or hydrological
signatures are useful to characterize the behavior of different components of river
discharge, from low flows, annual runoff to floods (Addor et al., 2018; McMillan et al.,
2017), and to assess the potential impact of climate change and human activities on
river regimes (Mahe et al., 2013). They can be used for various purposes, including
basin classifications, aquifer properties characterization, hydrological predictions in
ungauged catchments (Westerberg et al., 2016, Gnann et al., 2020) and to investigate
long term trends for different hydrological processes (Do et al., 2017; Nka et al., 2015).
We introduce here the African Dataset of Hydrometric Indices (ADHI) that aims at giving
access to an ensemble of hydrometric indices computed from an unprecedented large
ensemble of stations with daily discharge data (Tramblay et al., 2020, Tramblay and
Rouché, 2020). Thus, a minimum of useful information regarding the African rivers'
variability over the last 68 years can be shared with the international community, while
respecting the confidentiality of the original records when these are not allowed to be
publicly shared by the national authorities.

## 2. Data sources and processing

### 2.1 Data collection

The database used in the present work is based on the collection of stations from the Global Runoff Data Center (GRDC) and the SIEREM database (Boyer et al., 2006; Dieulin et al., 2019). The hydro-climatological data contained in SIEREM is the legacy from the former *Laboratoire d'Hydrologie* of the *Office de la recherche scientifique et technique outre-mer* (ORSTOM, now *Institut de Recherche pour le Développement*, IRD, France). It must be noted that in addition to the daily data, the SIEREM database also contains instantaneous rainfall and discharge for hundreds of experimental small catchments mostly established in the 1950s and 1960s. The criterion to include a station in the ADHI database is to have a minimum of 10 full years of daily discharge data between 1950 and 2018. Most of the hydrological stations in French-speaking countries have been set up and managed for decades by the ORSTOM Institute (Mahe and Olivry, 1999). At the time the data were processed, the SIEREM database included a total of 1046 series, with several of them being duplicates of the same monitoring station but for different time periods. There are a total of 101 stations with 2 times series, 42 stations with 3 time series, 24 stations with 4 time series and 7 stations with 5 time series. In most cases, one time series includes the longest record and that one was kept for the analysis in the present paper. For some stations, the different time series were differing substantially during the same period, due to different rating curves. A visual inspection of these series led to the elimination of erroneous or doubtful data. Only for 17 stations the time series were concatenated, after making sure the rating curve(s) applied on the different time periods to compute river discharge were adequate, by comparing daily runoff on a common period. Additionally, to these 1046 series, 933 stations have been retrieved from the GRDC database. For 106 of these stations, there was a duplicate station in the SIEREM database with longer time series and the latter were selected. After this data quality processing step, 672 stations were kept for SIEREM and 794 for the GRDC database for a total of 1466 stations (Figure 1). The stations from SIEREM mostly cover the Western, Central and Northern African regions and stations from the GRDC the Eastern and Southern parts of Africa. Figure 2 depicts the number of stations available per year, showing a sharp decline at the end of the 1980s, and shows the number of stations having from 10 to 69 years of record. It can be seen that, for about 100 stations, complete records are available over 50 years.

### 2.2 Data quality

Since the data collected are sometimes from manual records, they are subject to
possible errors in the reporting of discharge values. For outlier detection, no single
method can outperform visual inspection and local expert knowledge (Crochemore et
al., 2020). Indeed, in rivers with a strong variability in the annual regime and extremes,
the most important flood peaks may be wrongly reported as outliers. Consequently, we
carried out a visual inspection of the data when the maximum value was exceeding 5
times the median discharge. For only a few data points in the discharge time series,
some obvious errors were detected with daily discharge exceeding by several orders of
magnitude the median flow. In these cases, the data has been reported as missing data
in an absence of an objective criterion to correct the record. In addition, through visual
inspection it was possible to identify stations where some gap filling methods have been
applied (13 stations) or where the data are suspicious (28 stations). A flag has been
added in the metadata to identify these stations. It is worth noting that, for the stations of
the SIEREM database, most of the data were analyzed and criticized prior to the
inclusion in the database by the former ORSTOM hydrology laboratory, with therefore a
reduced level of error in the archived data.
In addition, to detect possible shifts in the data due to non-natural influences, such as
an artificial drift in the monitoring devices, changing instrumentation, recalibration of the
rating curve, or river regulation by dams or reservoirs, the Pettitt test (Pettitt, 1979) was
applied to mean annual runoff series. We reported the cases when the null hypothesis
of homogeneity was rejected, at the 5% significance level. 14 stations are reported with
homogeneity breaks in the metadata and this result was consistent with a visual
inspection. Since the possible causes of these changes in flow regime could be
manyfold and should be investigated with a more detailed case-by-case analysis, we
choose to keep these stations in the database, but to flag them accordingly.

**2.3 Climate characteristics**

This data collection results in the largest ever built database of daily discharge data in
Africa. These stations belong to different climate zones (Figure 1), according to the
Köppen-Geiger climate classification (Peel et al., 2007). The main climate zone
represented is Savannah (class Aw) for 687 stations corresponding to west and central
Africa basins. The second most represented climate zone is Steppe-hot (Bsh) for 207
stations located in the Sahel region and southern Africa (Botswana, Namibia). The
temperate with dry winter classes (Cwa and Cwb) include 187 and 125 stations,
respectively located in southern Africa (Zambia, Angola, Rwanda, Mozambique, South
Africa and Zimbabwe). The 98 stations belonging to the Desert-hot class (Bwh) are
mostly located in the northern and southern boundaries of the Sahara Desert. 87
Stations under a temperate climate with dry hot summer, corresponding to
Mediterranean climate (Csa) are found in North Africa and the southwestern part of
South Africa. Thus, the selected river basins are representative of most of the climate
zones in Africa. It must be noted that for large basins, such as the Congo, Niger or even
the Orange rivers, the climate type at the outlet may not be representative of the whole
catchment, that may span over diverse climate zones.
To document the mean annual precipitation and evapotranspiration at the catchment
scale, the CRU4 dataset has been considered (Harris et al., 2020). However, without
long-term and homogeneous ground monitoring networks over the African continent, no
best precipitation database could be identified for Africa as a whole (Sylla et al., 2013;
Beck et al., 2017; Awange et al., 2019; Satgé et al., 2020). For some regions, such as
Northern or Equatorial Africa, there are large differences between different remote
sensing or gauged-based precipitation products (Gehne et al., 2016; Harrison et al.,
2019; Nogueira, 2020), in particular for extreme precipitation events. This is the reason
why we choose to provide only mean annual precipitation, evapotranspiration and
temperature. This implies that the ADHI dataset does not provide metrics relying on
time series of precipitation or evapotranspiration, such as the runoff ratio, streamflow-
precipitation elasticity or catchment response time. To calculate these indices requiring
climatic time series for a given catchment, the user is advised to check first the best
available data for that area.
**2.4 Catchment delineation**
Station catchments areas have been delineated with the Hydroshed Digital Elevation
Model (DEM) at 15 sec resolution using the TopoToolbox2 algorithm (Schwanghart and
Scherler, 2014). The map of the catchments is shown in Figure 3. Despite a careful
check of the geographic coordinates of the stations, this type of automatic catchment
delineation procedure is prone to some errors, in particular in regions with low elevation
and flat terrain properties. This is particularly the case of catchments with endoreic
areas, such as the Niger, Chari and Logone basins, where the precision of the DEM is
crucial to identify these areas. Since the gauge locations are not necessarily located on
the streams derived from the DEM, The TopoToolbox2 makes possible to re-locate
automatically the gauges on the nearest river stream. However, this procedure did not
work for 61 catchments, with a catchment area error exceeding 10% compared to the
available metadata. For these basins, a manual procedure with the Arcmap® software
has been implemented to delineate the catchment boundaries from flow direction maps.
For several hundred of catchments, it is possible to compare the results of the
automatic delineation procedure with the catchment areas available in the SIEREM
database and the ORSTOM reports (available online at the adress:
https://horizon.documentation.ird.fr), which have been most often individually delineated
and carefully checked from ground knowledge over the years (Dieulin and Boyer, 2005).
For 37 stations in the SIEREM database, the catchment areas where not correct in the
metadata, by comparing the delineated catchments.
From the catchment delineated, the mean, maximum and minimum altitude from the
Hydroshed DEM have been extracted and included in the metadata. In addition, the
European Space Agency Climate Change Initiative Land cover data (ESA-CCI LC)
(ESA, 2017) has been extracted for each catchment for the year 2015. This database
contains land cover maps at a 300m spatial resolution for 38 classes, compliant with the
UN Land Cover Classification System (LCCS). The classes have been grouped into 8
new classes: forest, urban areas, cropland, irrigated croplands, grassland, shrubland,
sparse vegetation and bare land. Overall, the basins are characterized by a low
proportion of urban areas, a large proportion of forests, especially in the intertropical
zone (mean = 41%, median = 37%), and a majority of non-irrigated cultivated area, on
average covering 31% of the total area of the basins. Indeed, the irrigated crops
represent only 0.43% on average.
**2.5 River regulation**
Dams and reservoirs have also been extracted and added in the metadata of the
stations. The Global Reservoir and Dam Database (GRanD) v1.3 (Lehner et al., 2011)
has also been considered to identify regulated basins. The number of dams included in
each river basin has been extracted using the catchment boundaries. As shown in the
metadata of GRanD, most of the dams in Africa basins have been constructed around
the 1970s (Figure 4). The rivers could be considered regulated if at least one dam exists
in the catchment area, otherwise the river is considered natural (Figure 5). However, the
influence of dams and reservoirs on the flow regime are linked to the location of the
regulation structure, the portion of the basin controlled, and the management strategies.
For instance, in a large basin with only one dam located on a small headwater
catchment, its influence may not be distinguishable at the river outlet. On the other
hand, a station located immediately downstream a dam outlet may have its flow regime
strongly impacted by dam operations. It should be also noted that other regulation
structures like small dams or water diversion channels that may not be included in the
GRanD database could be present in the catchments considered natural (Lehner et al.,
2011; Pekel et al., 2016). This is particularly the case in semi-arid areas where earthen-
made channels, often informal, draw their water supply from the river itself, by building
small diverting structures (Underhill, 1984; Kimmage, 1991). They can represent a large
number of structures, but a variable amount of water withdrawal at the basin scale
(Barbier et al., 2009; Bouimouass et al., 2020). Similarly, no data is available yet on the
importance and impact of groundwater abstraction, if any, on the flow regime measured
at the stations.

# 3. Hydrometric indices


Here is presented the list of indices computed from daily discharge data. Most of the
indices are computed with the Toolbox for Streamflow Signatures in Hydrology
(TOSSH, available at the address: https://github.com/TOSSHtoolbox/) (Gnann et al.,
2021). The indices and signatures selected spans a large variety of runoff
characteristics from high to low flows, from previous literature (Poff et al., 1997; Richter
et al., 1996; Baker et al., 2004; Yadav et al., 2007; Clark et al., 2009; Estrany et al.,
2010; Sawicz et al., 2011; Euser et al., 2013; Safeeq et al., 2013; Addor et al., 2018;
McMillan, 2020).

### 3.1 Methodological considerations


Several signatures charactering baseflow rely on the application of a base flow filter.
Since the choice of the baseflow separation method can introduce uncertainties in the
calculation of these signatures (Su et al., 2016), two baseflow filtering methods are
compared: the Lyne and Hollick recursive digital filter (Ladson et al., 2013), with the
default values for the filter parameter (0.925) and the number of passes (3), and
alternatively the UKIH smoothed minima method (UKIH, 1980), that does not require
any calibration parameter. The base flow index (BFI) is the ratio between the baseflow
volume and the total streamflow volume. The baseflow recession (BaseflowR) is the
baseflow recession constant assuming an exponential recession behavior (Safeeq et
al., 2013). The base baseflow magnitude calculates the difference between the
minimum and the maximum of the baseflow regime, defined as the average baseflow on
each calendar day. The two base flow separation method compared to compute the
baseflow-related indices provide very similar results, with a correlation above 0.9 for all
indices obtained with the two approaches.

To compute the mean half flow date and the mean half flow interval, the beginning of
the hydrological year has been defined as the month following the month with the
minimum average runoff. Indeed, the hydrological year has different starting dates
across the African continent, in North Africa the hydrological year usually starts in
September, in western Africa around March-April and in January for southern Africa.
The mean half flow date is the day when the cumulative discharge reaches half of the
annual discharge. The mean half flow interval is the time span between: i) the date on
which the cumulative discharge since the start of water year reaches a quarter of the
annual discharge and ii) the date on which the cumulative discharge since the start of
water year reaches three quarters of annual discharge.
Some metrics are derived from the calculation of the Flow duration Curve (FDC), such
as its slope between the 33$^{rd}$ and 66$^{th}$ flow percentiles (McMillan et al., 2017), the peak
distribution, the slope between the 10$^{th}$ and the 50$^{th}$ percentiles of the FDC constructed
only with hydrographs peaks (Euser et al., 2013) and the variability index, the standard
deviation of the logarithms of discharge from 10$^{th}$ to the 90$^{th}$ percentiles of the FDC
(Estrany et al., 2010). It must be noted that 194 rivers have more than 50% of days with
zero-flow and for these stations, but also all the others with an intermittent regime,
several metrics derived from the Flow Duration Curve (FDC) are not adapted. For these
basins, specific methods to estimate the FDC should be applied (Rianna et al., 2013).
Similarly, there is no baseflow in these basins. Consequently, the indices relying to base
flow or the flow duration curve are not computed for these basins.
In addition, different hydrological signatures describing the hydrologic responses of the
basins are also provided. The flashiness index is defined as the sum of absolute
differences between consecutive daily flows (Baker et al., 2004), it reflects the
frequency and rapidity of short term changes in streamflow, especially during high runoff
events. The number of master recession curves (MRC) is computed from the changes
in recession slopes, and represent different reservoirs contributing to the runoff
response (Clark et al., 2009; Estrany et al., 2010). This signature can help to
understand the functional forms of storage–discharge relationships and identify model
structures adapted to represent it. The rising limb density is the ratio between the
number of rising limbs and the total amount of timesteps in the hydrograph (Sawicz et
al., 2011). It is a descriptor of the hydrograph shape and smoothness, without
consideration for the flow magnitude. Small values of the rising limb density indicate a
smooth hydrograph.
From the supplied indices, some other useful indicators could be derived. For example,
for hydrogeology applications it would be interesting to compute the low stage specific
discharge that is the ratio between the low-stage discharge and the area of the
watershed. This can be an indicator of aquifers' contribution to river discharge. The
main issue is related to the definition of the low-stage discharge. From the indices
proposed in the present database, it could be 5th percentiles of daily streamflow or the
minimum of 7-days consecutive streamflow, per year. Similarly, the low-flow index could
be computed from the ratio of the 90th and 50th percentiles of daily streamflow
(Smakhtin, 2001).
**3.2 Indices computed on the whole record**

These indices have been computed using the whole time series available for each
station. Consequently, they are computed on different base periods depending on the
stations, with the period of record for each station being made available in the
metadata. These indices include:

1. Mean daily streamflow, the arithmetic mean of daily data
2. Standard deviation of daily streamflow
3. Minimum daily streamflow
4. Maximum daily streamflow
5. Mean monthly streamflow (12 values from January to December)
6. 5th, 10th, 25th, 50th, 75th, 90th, 95th and 99th percentiles of daily streamflow
7. BFI_LH = Baseflow index, with the Lyne and Hollick baseflow separation method
8. BFI_UKIH = Baseflow index, with the UK Institute of Hydrology baseflow
408       separation method
9. BaseflowR = Baseflow recession
10. BaseflowM_LH = Baseflow magnitude, with the Lyne and Hollick baseflow
411       separation method
11. BaseflowM_UKIH = Baseflow magnitude, with the UK Institute of Hydrology
413       baseflow separation method
12. CoV = Coefficient of variation of runoff
13. HFD_mean = Mean half flow date
14. HFI_mean = Mean half flow interval
15. AC1 = lag-1 autocorrelation of flow
16. AC7 = lag-7 autocorrelation of flow
17. FDC_slop = Slope of flow duration curve
18. PeakDistri = Peak distribution
19. FlashI = Richards-Baker flashiness index
20. MRC_num = Number of master recession curves
21. Q_skew = Skewness of runoff
22. Q_var = Variance of runoff
23. RLD = Rising limb density
24. VariI = Variability index
25. Freq_0 = Frequency of zero-flow days

The basins included in the ADHI database include a wide range of catchment areas,
from a few square kilometers to several hundred thousand, in the case of large rivers
such as the Congo, Niger, Orange, Zambezi, Senegal, Okavango and Volta. As shown
in Figure 6, the average runoff is generally well correlated to the size of the basins with
nevertheless a variability linked to local climatic and geological conditions. The mean
annual precipitation is one of the explanatory factors of the observed ranges of mean
river runoff, but also strongly modulated by local conditions. A large number of basins
have an aridity index (ratio between precipitation and potential evapotranspiration) of
less than 0.60, indicative of arid to semi-arid conditions (figure 7a). The varying degrees
of aridity encountered in the basins are an important explanatory factor for the
hydrological response at the African scale. For instance, the coefficient of variation of
runoff (figure 7b) or the flashiness index (figure 7c) have greater values under
conditions of increasing aridity.

**3.3 Indices computed on monthly or annual basis**

These indices have been computed for each calendar year, for consistency with other
databases such as GSIM (Do et al., 2018; Gudmundsson et al., 2018). These indices
have been computed for the years with less than 5% missing data:

1. Mean annual runoff
2. Minimum of 7-days consecutive streamflow, per year, and corresponding date
3. Annual maximum runoff, and the corresponding date
4. Annual values for the 5th, 10th, 25th, 50th 75th, 90th, 95th and 99th percentiles of daily streamflow

In addition to these annual series, the monthly time series contains for each month the
mean, maximum and minimum runoff, the last column being the number of missing
days per month. There is one file per station. It is advised to consider the monthly
values only for the months with no missing values, or missing values less than 10% or
459  5%.

These time series make it possible to analyze the long-term evolution of mean and
extreme runoff (Tramblay et al., 2020), but can also be useful to validate hydrological
modelling results. Focusing on extreme high and low runoff, very different seasonal
patterns of occurrence could be observed for different regions of Africa. On figure 8 are
plotted the mean dates of annual maximum runoff and the annual minimum of 7-day
runoff. This seasonal analysis has been performed with directional statistics (Burn,
1997; Mardia et al., 2015): the dates of occurrence were converted into angular values
to compute the mean date of occurrence ($\theta$) together with the concentration index ($r$),
which is a measure of the flood occurrences variability around the mean date. The
annual maximum runoff shows three distinct patterns (Figure 8): First, stations with
floods occurring during December-February in northern and southern Africa, with a
strong variability of their date of occurrence. Second, the stations in western Africa with
floods occurring during summer and a low seasonal variability. Third, the stations in
central-south Africa, with floods occurring in boreal spring and early summer with
various degrees of variability depending on the sub-region considered and the level of
aridity. For annual minimum runoff, the patterns are usually reversed, with the low flow
period spanning on average during June to October in North Africa, January-March in
western Africa, and between September and November in southeast Africa. Yet this
global picture hides local behaviors such East-West contrast in southern Africa or the
North-South gradient in West Africa (Mahe et al., 2013). Similarly, the observed
variability even for some neighboring catchments reflects the local influences of
topography, soils and land cover. As noted previously, the seasonal variability of
extreme high or low runoff events is also strongly related to the catchment aridity.

## 4. Data availability

The ADHI database is available for download at: https://doi.org/10.23708/LXGXQ9
(Tramblay and Rouché, 2020). These different files are supplied in the AHDI database:

The ADHI_stations.dat file contains:

-Unique identifier for each station
-Station code (native code from the original datasource)
-Station Name
-Data Source
-Catchment Area (km²)
-Mean Altitude (m)
-Maximum Altitude (m)
-Minimum Altitude (m)
-Mean annual precipitation (mm)
-Mean annual evapotranspiration (mm)
-Mean annual temperature (°C)
-Forest cover (%)
-Urban areas (%)
-Cropland (%)
-Cropland, irrigated (%)
-Grassland (%)
-Shrubland (%)
-Sparse vegetation (%)
-Bare land (%)
-Starting year of the data records
-Ending year of the data records
-Longitude (WGS84)
-Latitude (WGS84)
-Number of dams
-Country
-Flag, 0: no identified data issue, 1: some gap filling detected, 2: suspicious data, 3:
Obvious regime break
The ADHI_summary.dat file contains for each station (lines) the following variables
(columns):
Mean_q = Mean daily streamflow (m3/s)
Std_q = Standard deviation of daily streamflow
Mini_q = Minimum daily streamflow
Maxi_q = Maximum daily streamflow
Jan_q, Fev_q... Dec_q = Mean monthly streamflow (12 values from January to
December)
q5th, q10th, q25th, q50th q75th, q90th, q95th and q99th percentiles of daily streamflow
BFI_LH = Baseflow index, with the Lyne and Hollick baseflow separation method
BFI_UKIH = Baseflow index, with the UK Institute of Hydrology baseflow separation
method
BaseflowR = Baseflow recession
BaseflowM_LH = Baseflow magnitude, with the Lyne and Hollick baseflow separation
method
BaseflowM_UKIH = Baseflow magnitude, with the UK Institute of Hydrology baseflow
separation method
CoV = Coefficient of variation of runoff
HFD_mean = Mean half flow date
HFI_mean = Mean half flow interval
AC1 = lag-1 autocorrelation of flow
AC7 = lag-7 autocorrelation of flow
FDC_slop = Slope of flow duration curve
PeakDistri = Peak distribution
FlashI = Richards-Baker flashiness index
MRC_num = Number of master recession curves
Q_skew = Skewness of runoff
Q_var = Variance of runoff
RLD = Rising limb density
VariI = Variability index
Freq_0 = Frequency of zero-flow days

The compressed folders AnnualMean.zip, AnnualMax.zip, Annual7DayMin.zip,
AnnualPercentiles.zip contains time series for mean annual runoff, annual maximum
runoff, annual minimum of 7-day discharge and annual values for the 5th, 10th, 25th,
50th 75th, 90th, 95th and 99th percentiles of daily streamflow. There is one file per
station.

The compressed folder MonthlySeries.zip contains for each month the mean, maximum
and minimum runoff, the last column is the numer of missing days per month. There is
one file per station.

The compressed folder Plots.zip contains for each station a plot of the daily discharge
data available.

The compressed folder Catchment_boundaries.zip contains the catchment boundaries
in the shapefile format (one .shp file per basin).

The compressed folder Catchment_plots.zip contains for each basin a plot of the
catchment area in .PNG format.

# 5. Conclusions and perspectives

This new hydrological database brings together the largest number of African river flow
measurement stations, in comparison with other previously published datasets. In this
ADHI dataset, we included a total of 1466 stations with at least 10 years of discharge
data between 1950 and 2018, for a mean record length of 33.3 years. Half of the
stations have more than 30 years of data. By comparison, the recent GSIM database
contains 979 stations in Africa, with a record length varying from 1 year to 110 years
until 2015, and a mean record length of 33.8 years. This ADHI database results from a
pooling of the GRDC and SIEREM databases, built from contributions of several
agencies in African countries in charge of the management of hydrological
measurement networks. This database will be regularly updated with data from SIEREM
and GRDC. Since most of the pre-processing steps have been automated, it would be
possible to increase the number of stations considered or the length of the data series,
if more data would become available. The data from the SIEREM database is already
regularly updated from contributions of different institutes. In the future, individual
contributions from researchers or institutes will be also welcome to increase the spatio-
temporal coverage of the data. The FRIEND program (UNESCO/IHP) will also
contribute to increase the number of stations through coordinated efforts at the regional
level. The dataset provides a series of indices that describes a wide range of mean and
extreme runoff properties, allowing the characterization of the hydrological regime and
applications linked to the management of water resources and hydrological risks. This
database includes different catchment sizes and rivers with different hydrological
regimes that makes possible to analyze the behavior of rivers in very different contexts
for a wide range of scales.

More broadly, this ADHI database could contribute to a better knowledge on African
hydrology. For instance, the impacts of dams on river discharge remains largely
unquantified at the scale of Africa (Biemans et al., 2011). From these indices, various
applications can be sought. For example, the percentiles of the daily streamflow could
be useful to calibrate hydrological models using the flow duration curve (McMillan et al.,
2017) and to constrain model outputs (Tumbo and Hughes, 2015; Ndzabandzaba and
Hughes, 2017). Flow duration curves are also useful for catchment classification
according to their rainfall-runoff response (Cheng et al., 2012). In the recent years,
global runoff simulations have been provided by the Global Flow Awareness System,
with land surface or global hydrological model driven by reanalysis data (Alfieri et al.,
2020; Harrigan et al., 2020). Yet, due to the small number of stations representing
African basins in the currently available databases preventing a robust calibration of the
models, the hydrological simulations have a poor performance (Harrigan et al., 2020).
More generally, this new ADHI database could open perspectives to apply hydrological
models in African basins, in particular combined with recent remote sensing data
products (Brocca et al., 2019; Satgé et al., 2020). Beside deterministic hydrological
modelling approaches, several statistical methods to estimate the return levels of floods
have been proposed, in order to safely design dams, reservoirs, sewers or other water
regulation structures. Regional frequency analysis methods have been applied to
estimate floods in ungauged basins in several African countries such as Morocco (Zkhiri
et al., 2017), Tunisia (Ellouze and Abida, 2008), South Africa (Nathanael et al., 2018;
Smakhtin et al., 1997), or the Volta basin (Komi et al., 2016). However studies at a
larger regional scale remain very scarce (Farquharson et al., 1992; Padi et al., 2011)
while there is a strong need to improve the knowledge on hydrological hazards in
African countries (Di Baldassarre et al., 2010). With this recent database becoming
available, it could be possible to develop regional frequency analysis techniques for
floods or low flows tailored for the African context, taking also into account the impacts
of global changes.

**Acknowledgements**

River runoff has been obtained from The Global Runoff Data Centre, 56068 Koblenz,
Germany and included in the database with their authorization. We would like to thank
the GRDC (https://www.bafg.de/GRDC/) for granting access to their data. A large part of
the data processed in the present study comes from the SIEREM database
(http://www.hydrosciences.fr/sierem), and the authors wish to express their gratitude to
all the persons who contributed to this database over the years. The authors wish to
thank Sebastian Gnann for his assistance and feedback with the TOSSH toolbox
(https://github.com/TOSSHtoolbox/).
The database is available from the online repository: https://doi.org/10.23708/LXGXQ9
Additional indices could be computed upon reasonable request to the corresponding
author.
This work is dedicated to the memory of Claudine Dieulin who passed away in January
2020 during the course of this project
**New references**
Addor, N., Nearing, G., Prieto, C., Newman, A. J., Le Vine, N. and Clark, M. P.: A
Ranking of Hydrological Signatures Based on Their Predictability in Space, Water
Resour. Res., 54(11), 8792–8812, https://doi.org/10.1029/2018WR022606, 2018.
Awange, J. L., Hu, K. X. and Khaki, M.: The newly merged satellite remotely sensed,
gauge and reanalysis-based Multi-Source Weighted-Ensemble Precipitation: Evaluation
over Australia and Africa (1981–2016), Science of The Total Environment, 670, 448–
465, https://doi.org/10.1016/j.scitotenv.2019.03.148, 2019.
Baker, D. B., Richards, R. P., Loftus, T. T. and Kramer, J. W.: A new flashiness index:
characteristics and applications to midwestern rivers and streams, J Am Water
Resources Assoc, 40(2), 503–522, https://doi.org/10.1111/j.1752-1688.2004.tb01046.x,
662 2004.

Beck, H. E., Vergopolan, N., Pan, M., Levizzani, V., van Dijk, A. I. J. M., Weedon, G. P.,
Brocca, L., Pappenberger, F., Huffman, G. J. and Wood, E. F.: Global-scale evaluation
of 22 precipitation datasets using gauge observations and hydrological modeling,
Hydrol. Earth Syst. Sci., 21(12), 6201–6217, https://doi.org/10.5194/hess-21-6201-
667 2017, 2017.

Burn, D. H.: Catchment similarity for regional flood frequency analysis using seasonality
measures, Journal of Hydrology, 202(1–4), 212–230, https://doi.org/10.1016/S0022-
670 1694(97)00068-1, 1997.

Clark, M. P., Rupp, D. E., Woods, R. A., Tromp-van Meerveld, H. J., Peters, N. E. and
Freer, J. E.: Consistency between hydrological models and field observations: linking
processes at the hillslope scale to hydrological responses at the watershed scale,
Hydrol. Process., 23(2), 311–319, https://doi.org/10.1002/hyp.7154, 2009.
ESA: Land Cover CCI Product User Guide Version 2. Tech. Rep.,
maps.elie.ucl.ac.be/CCI/viewer/download/ESACCI-LC-Ph2-PUGv2_2.0.pdf, 2017.
Estrany, J., Garcia, C. and Batalla, R. J.: Hydrological response of a small
mediterranean agricultural catchment, Journal of Hydrology, 380(1–2), 180–190,
https://doi.org/10.1016/j.jhydrol.2009.10.035, 2010.
Euser, T., Winsemius, H. C., Hrachowitz, M., Fenicia, F., Uhlenbrook, S. and Savenije,
H. H. G.: A framework to assess the realism of model structures using hydrological
signatures, Hydrol. Earth Syst. Sci., 17(5), 1893–1912, https://doi.org/10.5194/hess-17-
683 1893-2013, 2013.

Gehne, M., Hamill, T. M., Kiladis, G. N. and Trenberth, K. E.: Comparison of Global
Precipitation Estimates across a Range of Temporal and Spatial Scales, J. Climate,
29(21), 7773–7795, https://doi.org/10.1175/JCLI-D-15-0618.1, 2016.
Gnann, S. J., Coxon, G., Woods, R. A., Howden, N. J. K. and McMillan, H. K.: TOSSH:
Pre-release, Zenodo., 2021.
Harris, I., Osborn, T. J., Jones, P. and Lister, D.: Version 4 of the CRU TS monthly high-
resolution gridded multivariate climate dataset, Sci Data, 7(1), 109,
https://doi.org/10.1038/s41597-020-0453-3, 2020.
Harrison, L., Funk, C. and Peterson, P.: Identifying changing precipitation extremes in
Sub-Saharan Africa with gauge and satellite products, Environ. Res. Lett., 14(8),
085007, https://doi.org/10.1088/1748-9326/ab2cae, 2019.
Ladson, A. R., Brown, R., Neal, B. and Nathan, R.: A Standard Approach to Baseflow
Separation Using The Lyne and Hollick Filter, Australasian Journal of Water Resources,
17(1), 25–34, https://doi.org/10.7158/13241583.2013.11465417, 2013.
Mahe, G., Lienou, G., Descroix, L., Bamba, F., Paturel, J. E., Laraque, A., Meddi, M.,
Habaieb, H., Adeaga, O., Dieulin, C., Chahnez Kotti, F. and Khomsi, K.: The rivers of
Africa: witness of climate change and human impact on the environment, Hydrol.
Process., 27(15), 2105–2114, https://doi.org/10.1002/hyp.9813, 2013.
Mardia, K. V., Birnbaum, Z. W. and Lukacs, E.: Statistics of Directional Data., Elsevier
Science, Saint Louis. http://qut.eblib.com.au/patron/FullRecord.aspx?p=1901433, last
access: 21 January 2021, 2015.
McMillan, H.: Linking hydrologic signatures to hydrologic processes: A review,
Hydrological Processes, 34(6), 1393–1409, https://doi.org/10.1002/hyp.13632, 2020.
McMillan, H., Westerberg, I. and Branger, F.: Five guidelines for selecting hydrological
signatures, Hydrological Processes, 31(26), 4757–4761,
https://doi.org/10.1002/hyp.11300, 2017.
Nogueira, M.: Inter-comparison of ERA-5, ERA-interim and GPCP rainfall over the last
40 years: Process-based analysis of systematic and random differences, Journal of
Hydrology, 583, 124632, https://doi.org/10.1016/j.jhydrol.2020.124632, 2020.
Poff, N. L., Allan, J. D., Bain, M. B., Karr, J. R., Prestegaard, K. L., Richter, B. D.,
Sparks, R. E. and Stromberg, J. C.: The Natural Flow Regime, BioScience, 47(11),
769–784, https://doi.org/10.2307/1313099, 1997.
Rianna, M., Efstratiadis, A., Russo, F., Napolitano, F. and Koutsoyiannis, D.: A
stochastic index method for calculating annual flow duration curves in intermittent rivers,
Irrig. and Drain., 62(S2), 41–49, https://doi.org/10.1002/ird.1803, 2013.
Richter, B. D., Baumgartner, J. V., Powell, J. and Braun, D. P.: A Method for Assessing
Hydrologic Alteration within Ecosystems, Conservation Biology, 10(4), 1163–1174,
https://doi.org/10.1046/j.1523-1739.1996.10041163.x, 1996.
Safeeq, M., Grant, G. E., Lewis, S. L. and Tague, Christina. L.: Coupling snowpack and
groundwater dynamics to interpret historical streamflow trends in the western United
States, Hydrological Processes, 27(5), 655–668, https://doi.org/10.1002/hyp.9628,
725 2013.

Satgé, F., Defrance, D., Sultan, B., Bonnet, M.-P., Seyler, F., Rouché, N., Pierron, F.
and Paturel, J.-E.: Evaluation of 23 gridded precipitation datasets across West Africa,
Journal of Hydrology, 581, 124412, https://doi.org/10.1016/j.jhydrol.2019.124412, 2020.
Sawicz, K., Wagener, T., Sivapalan, M., Troch, P. A. and Carrillo, G.: Catchment
classification: empirical analysis of hydrologic similarity based on catchment function in
the eastern USA, Hydrol. Earth Syst. Sci., 15(9), 2895–2911,
https://doi.org/10.5194/hess-15-2895-2011, 2011.
Su, C.-H., Costelloe, J. F., Peterson, T. J. and Western, A. W.: On the structural
limitations of recursive digital filters for base flow estimation: EVALUATION OF
GENERALIZED BASE FLOW SEPARATION FILTERS, Water Resour. Res., 52(6),
4745–4764, https://doi.org/10.1002/2015WR018067, 2016.
Sylla, M. B., Giorgi, F., Coppola, E. and Mariotti, L.: Uncertainties in daily rainfall over
Africa: assessment of gridded observation products and evaluation of a regional climate
model simulation, Int. J. Climatol., 33(7), 1805–1817, https://doi.org/10.1002/joc.3551,
740 2013.

Tramblay, Y., Villarini, G. and Zhang, W.: Observed changes in flood hazard in Africa,
Environ. Res. Lett., 15(10), 1040b5, https://doi.org/10.1088/1748-9326/abb90b, 2020.
UKIH: Low Flow Studies Reports, 1980.
Yadav, M., Wagener, T. and Gupta, H.: Regionalization of constraints on expected
watershed response behavior for improved predictions in ungauged basins, Advances
in Water Resources, 30(8), 1756–1774,
https://doi.org/10.1016/j.advwatres.2007.01.005, 2007.

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

**Figures**

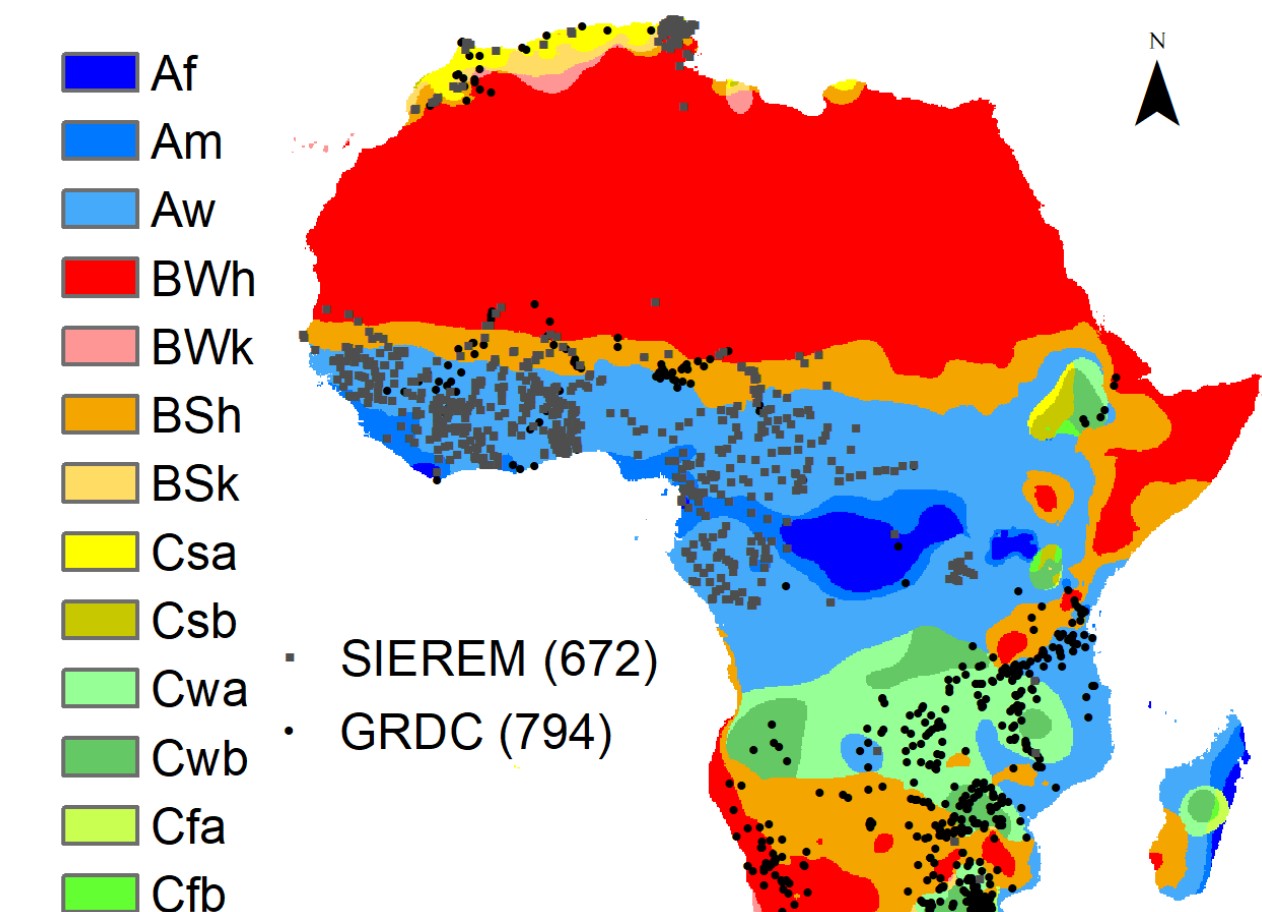

Af
Am
Aw
BWh
BWk
BSh
BSk
Csa
Csb
Cwa
Cwb
Cfa
Cfb

SIEREM (672)
GRDC (794)

0    650  1 300        2 600 Km

Figure 1: Map of the selected stations for the ADHI database from the SIEREM and
GRDC datasets. The different colors represent the main climate zones in Africa from the
1004           Köppen-Geiger climate classification (Peel et al., 2007)


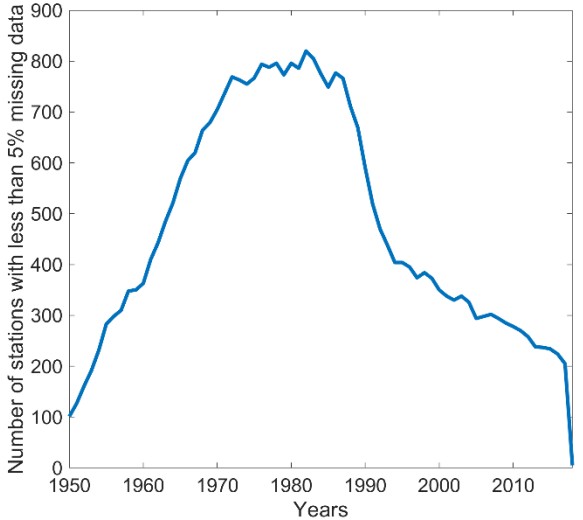 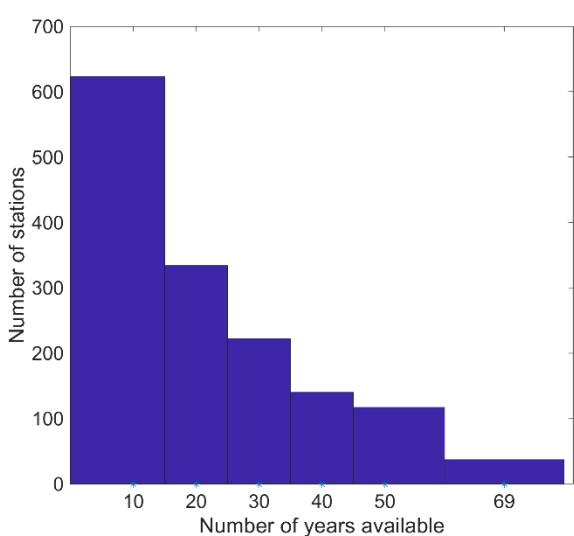


Figure 2: Number of available stations per year with less than 5% missing data (left) and
number of stations available for different record lengths (right)

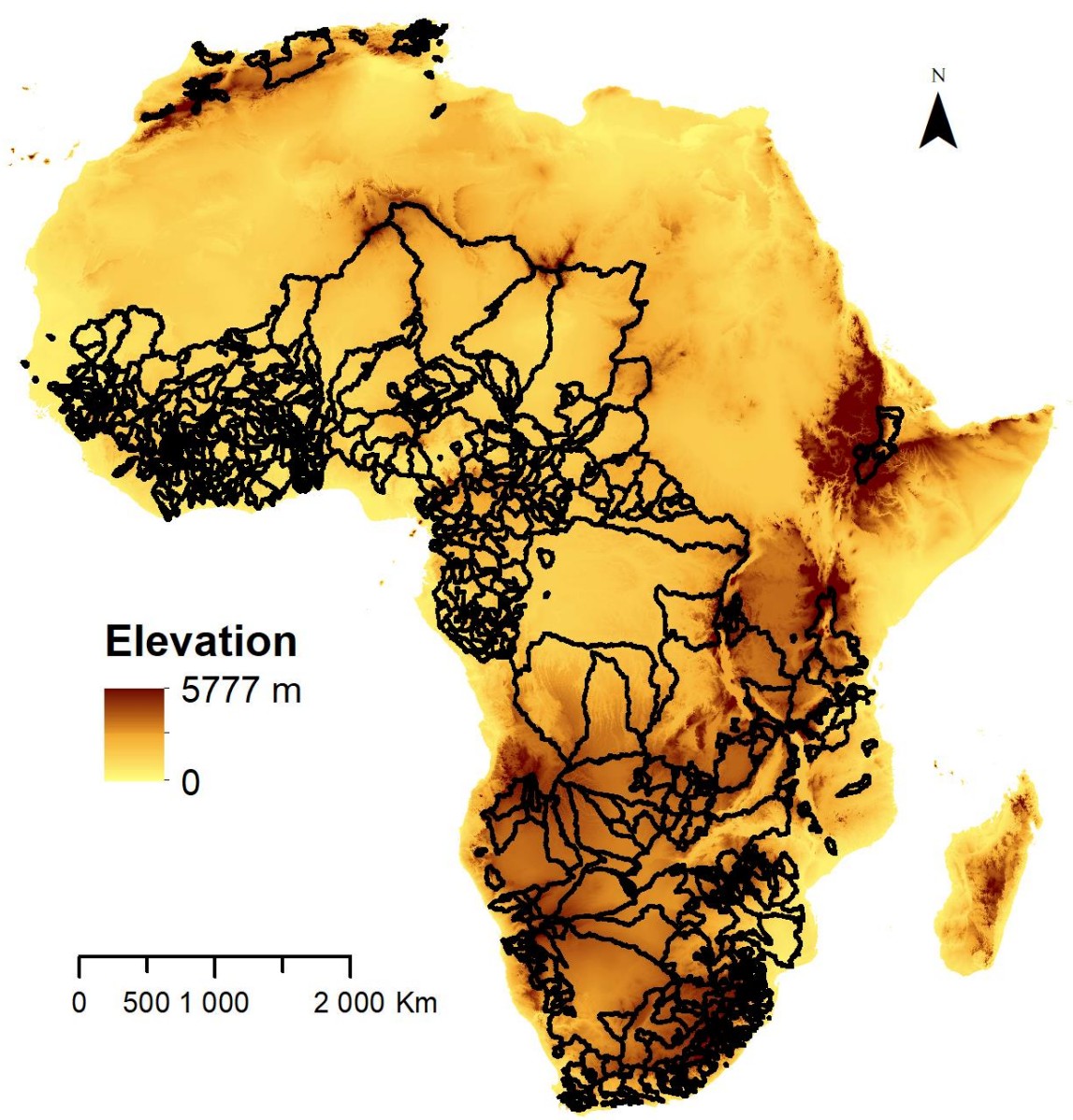

Figure 3: Map of the delineated catchment boundaries in black, with elevation from HydroSheds digital elevation model (https://www.hydrosheds.org/).

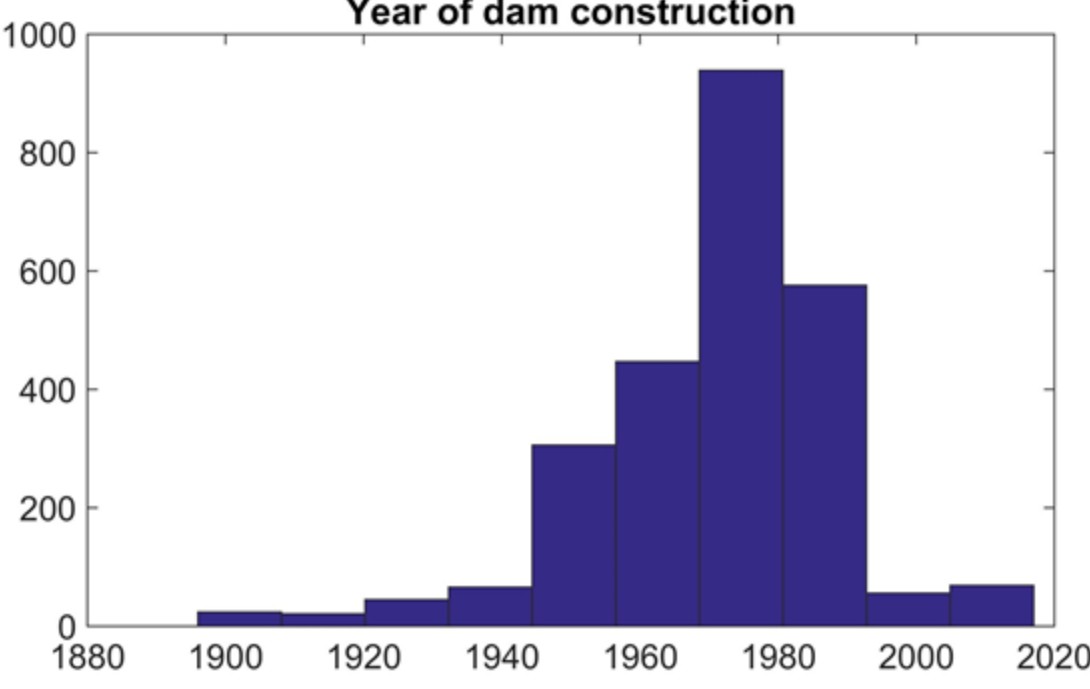


Figure 4: Years of building date for dams located in the catchment database (data from
the Global Reservoir and Dam Database v1.3)

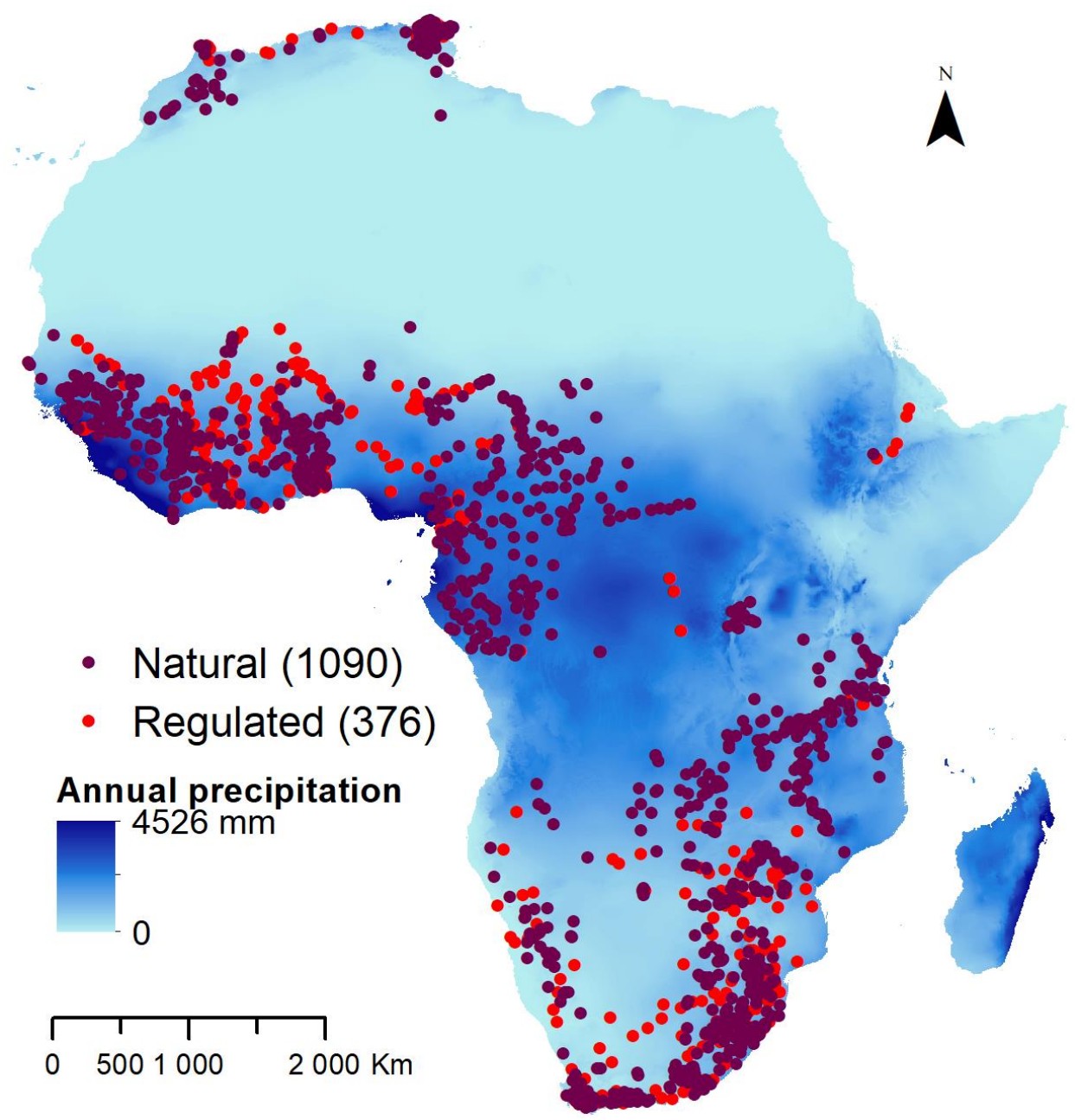

Figure 5: Map of stations with a natural or regulated flow regime. Basins are considered regulated if they contain at least one dam or reservoir from the GRanD database (Lehner et al., 2011). Mean annual precipitation between 1970 and 2000 is provided from the WorldClim database (Fick and Hijmans, 2017).

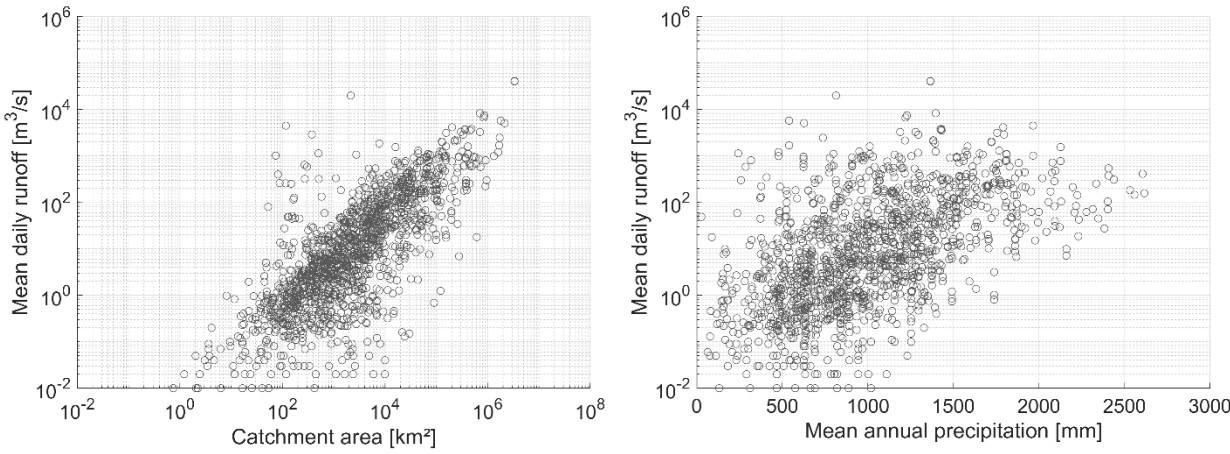

Figure 6: Relationship between mean daily river discharge and catchment area (left)
and mean annual precipitation (right)

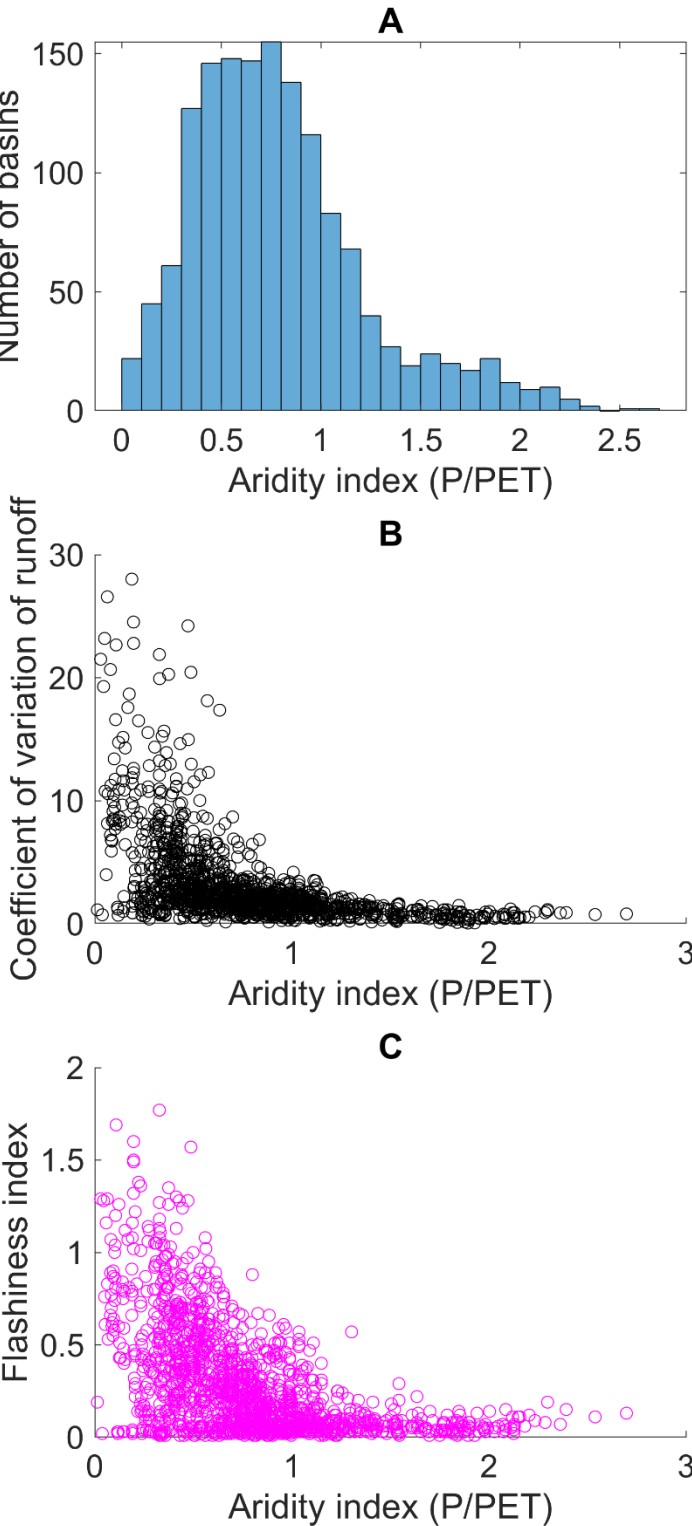

Figure 7: histogram of the aridity index per basin (A), relationship between the aridity
index and the coefficient of variation of runoff (B), relationship between the aridity index
1067                        and the flashiness index (C)


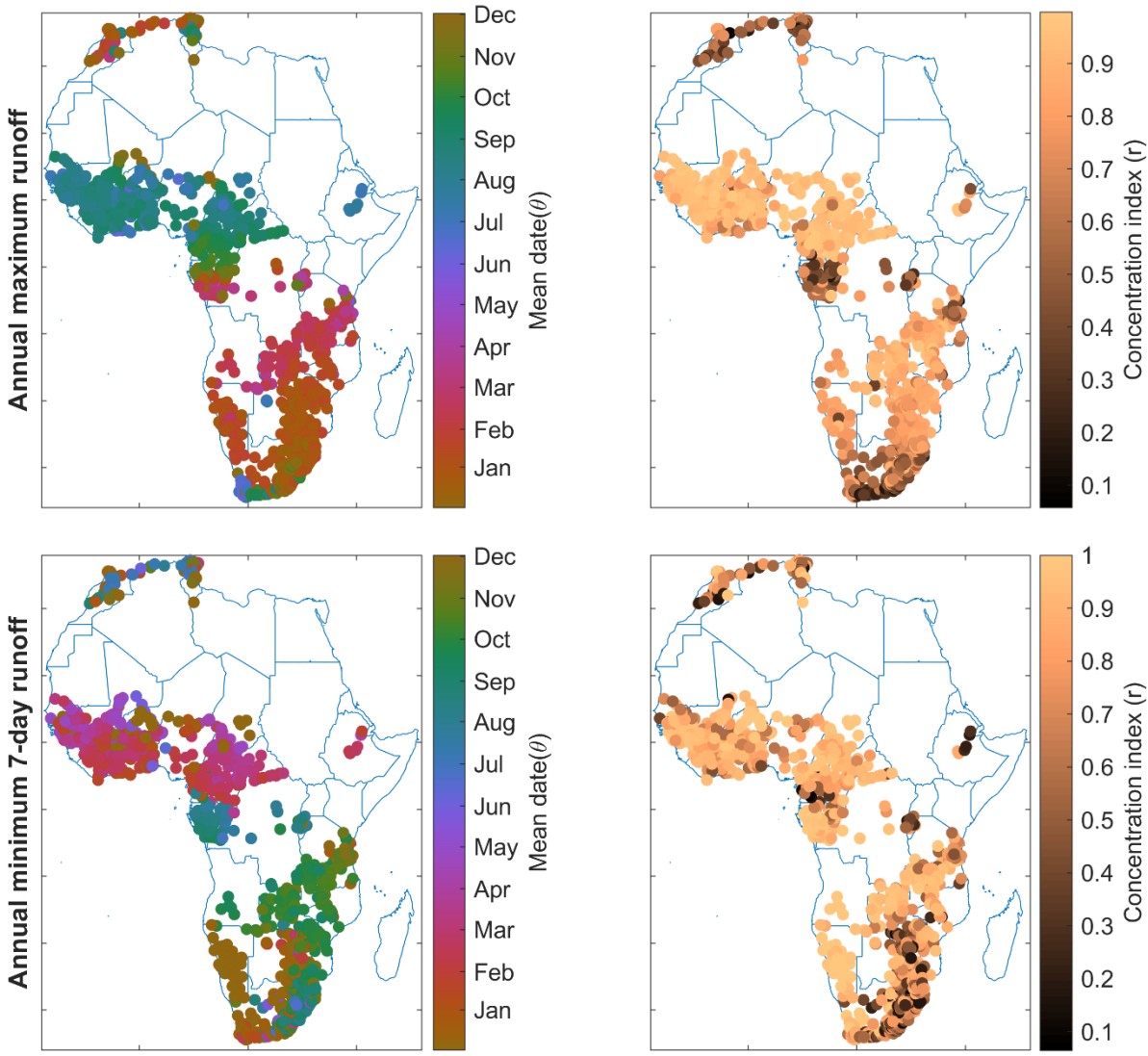

Figure 8: Mean date of occurrence (left) of annual maximum runoff and annual minimum
of 7-day runoff, together with the variability around the mean date (right) represented by
1073                              the concentration index