# Peer review of "ADHI: The African Database of Hydrometric Indices 1 (1950-2018) 2"

_Earth System Science Data, 2020_

## Referee Comment (RC1) · Anonymous Referee #1 · 4 Dec 2020

Review Tramblay et al. 2020 ESSD

Tramblay et al. present the African Database of Hydrometric Indices (ADHI), a database containing streamflow metrics and metadata for a large sample of African catchments. They describe how the database was created, what it contains, and how the database can be accessed. They also provide some background information and discuss potential uses of the database.

The paper is well organised and mostly well written. The idea of providing a large hydrometric database for Africa is a very welcome one and within the scope of ESSD. While I think that the paper (and the database) could be published with only minor revisions, I am also left with the feeling that the database could be made more attractive by expanding it (more signatures, more metadata, etc.). I leave it up to the authors to

decide whether to expand the dataset or not, but below I provide some comments on why I think this would be helpful.

Major comments:

The title is clear, but when I first read the abstract, I thought that this database also contains streamflow time series. I think it should be made clearer in the abstract that the database does not contain any streamflow time series. I am curious to hear about the authors' experiences with the data owners. What are the main reasons for not allowing to share the (raw) data? This does not have to be part of the paper, but I would be curious to know.

I think providing more hydrological metrics/signatures would make the database more attractive. Most of the metrics provided are statistical metrics, with somewhat limited use for hydrological (process) studies. There is a wide range of potentially more meaningful signatures (e.g. Addor et al., 2018; or see McMillan, 2020, for a review focusing on process-based signatures).

Similarly, while the database contains some metadata, there is potential to provide much more information. You already calculated catchment shapefiles, which could be shared as well. You could also use the shapefiles to extract more catchment characteristics from global (or African) data products. For example, there are products for P, T, PET, and for many catchment attributes, which would make the database more attractive (in my opinion). Otherwise it might be a bit hard to compete with recently published datasets that provide time series and various catchment attributes, despite the geographically unique coverage of your database.

Since the original time series are not accessible, there is no way to reproduce the derived indices. But I think that it would be helpful if you could share the code used to create the database.

It would be helpful to discuss a little bit more quantitatively how this database differs for

example from GSIM. How many more catchments are in there? This will make it easier for users to see the advantages of that dataset.

There are a few language errors and some unclear sentences. It's mostly minor, but I think proofreading the paper again would be helpful. I made some comments on that in the list below.

Minor comments:

L:85-86: "carefully checked for quality control". I would rephrase this – you do a quality control, that is, you check for quality. But you don't check for quality control.

L.86-87: This first suggested to me that the database also includes the discharge data, which (unfortunately) it doesn't. I would suggest rephrasing it so that this is clear. The same holds true for the following sentences.

L.104: "aquifers" – remove the "s"

L.126: You mentioned two main reasons, but you then list 3 points.

L.134: I would suggest writing 20th century instead of using Roman numerals.

L.132-136: That sentence is quite long and could be simplified/split.

L.144: The reference Gnann et al. (2020) doesn't really fit here as the study does not look at climate change or human activities. It would fit better into the next sentence after Westerberg et al. (2020).

L.181: "A careful inspection" – please be more explicit here.

L.187: "After this data quality processing step,…". Do you mean all the steps described above, that is the minimum length requirement, checking for duplicate time series, and merging with the GRDC? L.197: Replace "About" with some other word, it sounds a bit awkward.

L.202-205: I would suggest rephrasing that sentence. I think I understand what you

want to say here, but it's a bit unclear.

L.219-220: "manyfold" instead of "many folds"

L.230 and others: perhaps write southern Africa to clearly distinguish between the south of Africa and South Africa. Technically, the capitalisation should suffice, but it can be a bit confusing.

L.231 "includes" remove the "s"

L.283: I'm not sure what you mean by "relative indices computed with a base period as reference, such as standardized drought indices." To which indices do you refer to?

L.309: Here you use "indexes", earlier on "indices"

L.316: I am not entirely sure about the purpose of this paragraph. It reads like a discussion, which I didn't expect in that section. But it's a bit vague and doesn't really help the reader (in my opinion).

L.351: "sumary.txt file" is called "summary.tab" in the database

L.360: "contains" – remove the "s"

L.380: do you mean "organisations" here rather than "organisms"?

L. 418-420: Could you provide links (if they exist) to the GRDC and the SIEREM database websites? (Obviously the data cannot be accessed that way, but providing contact info of the data owners would be helpful.)

Figure 1: In the caption you mention twice that regulated basins are basins with dams.

References

Addor, N., Nearing, G., Prieto, C., Newman, A.J., Le Vine, N. and Clark, M.P., 2018. A ranking of hydrological signatures based on their predictability in space. Water Resources Research, 54(11), pp.8792-8812.

[Figure]

McMillan, H., 2020. Linking hydrologic signatures to hydrologic processes: A review. Hydrological Processes, 34(6), pp.1393-1409.

---

## Referee Comment (RC2) · Hong Do (Referee) · 17 Dec 2020

This manuscript introduces the African Database of Hydrometric Indices (ADHI), an unprecedented collection of streamflow signatures for Africa. I believe the data product will be greatly appreciated by the regional and global hydrology communities (myself included) as it can potentially fill a significant gap in in-situ records of streamflow and thus can advance hydrologic research over the tropics.

Although I fully support the publication of this dataset, I have the feeling that the dataset (and associated manuscript) has been developed in a rush, and thus has missed an opportunity to become a great product that benefits a broader range of users. For example, only time series of annual mean, annual max, and annual 7-d min streamflow

indices are published although the authors have done an excellent job in synthesizing and quality-controlling that much daily streamflow data.

I'm recommending a major revision to encourage the authors to improve and make the dataset more attractive to the international community. Below, I listed three major improvements that I strongly suggest the authors consider.

1. Expand the streamflow indices that could be accessed publicly.

- I appreciate the challenges in data restriction that the authors could have faced, but I think that it is defensible to increase the number of published time-series indices from three (in the current phase) to that described in Gudmundsson et al. (2018) - which I believe the authors have mentioned in their manuscript.

- I do think that at the minimum, time series of monthly indices (mean, max, min) would be highly appreciated by the global community to support a wide range of hydro-climatological research.

- I feel that the static percentiles that currently published in the summary text file could be re-processed using the block-window approach (e.g. yearly) to derive time-series that are useful to assess hydrological changes in Africa.

2. Providing catchment shapefiles

Figure 3 shows that the authors have also compiled/generated a great collection of catchment boundaries. This is another great asset that could benefit a broad range of end-users. I think publishing this information will not inflict any troubles regarding data policies.

3. Although I have not provided any specific comments on the manuscript (as I expect a major revision to make the manuscript become stronger), I have some general comments on the writing that may help better highlight the contribution of this dataset:

- The title: please consider some assertion titles such as "The production of seventyyear long streamflow indices for 1500 stations across Africa." This type of title reflects better the usefulness of the AHDI and thus will be more attractive to prospective users.

- Some figures were not associated with an insightful discussion (Figure 1 is not exactly what described in Section 2.1; Figure 5 was completely left out in the discussion). Please expand your discussion regarding any "lesson-learned" working with this dataset. For instance, some discussion about the relationship between the annual precipitation (shown as the background of Figure 1) and annual streamflow (generated by the authors) could be useful; section 4 contains effectively only two lists of bullet points - but could be expanded to include examples of "spurious patterns", substantial local changes, or improvement relative to the GRDC (see below.)

- I also think a map showing improvement of ADHI relative to the GRDC database (perhaps in Section 4) could be useful for end-users. For instance, the authors can classify stations into three categories (i) new stations (relative to GRDC), (ii) extended record stations, and (iii) no improvement.

The efforts of the authors to publish this data are greatly commended, and I am very excited about the release of the updated AHDI.

---

## Author Comment (AC1) · 13 Jan 2021

**Tramblay et al. present the African Database of Hydrometric Indices (ADHI), a database containing streamflow metrics and metadata for a large sample of African catchments. They describe how the database was created, what it contains, and how the database can be accessed.**

**They also provide some background information and discuss potential uses of the database. The paper is well organised and mostly well written. The idea of providing a large hydrometric database for Africa is a very welcome one and within the scope of ESSD. While I think that the paper (and the database) could be published with only minor revisions, I am also left with the feeling that the database could be made more attractive by expanding it (more signatures, more metadata, etc.). I leave it up to the authors to decide whether to expand the dataset or not, but below I provide some comments on why I think this would be helpful.**

We would like to thank you for this positive evaluation of the database and the corresponding data paper. We increased the number of hydrological signatures present in the database, by using the TOSHH toolbox recently released (https://sebastiangnann.github.io/TOSSH_development/p2_signatures.html).

We also provided the catchment boundaries and additional attributes such as land cover and mean precipitation and evapotranspiration. These new files will appear shortly in the online repository.

**Major comments:**

**The title is clear, but when I first read the abstract, I thought that this database also contains streamflow time series. I think it should be made clearer in the abstract that the database does not contain any streamflow time series.**

We modified the abstract and added "This new African Database of Hydrometric Indices (ADHI) provides a wide range of hydrometric indices and hydrological signatures computed from different sources of data carefully checked for quality control".

**I am curious to hear about the authors' experiences with the data owners. What are the main reasons for not allowing to share the (raw) data? This does not have to be part of the paper, but I would be curious to know.**

Different problems can be encountered for data access. Frequently, the organizations in charge of collecting and monitoring hydrological measurements do not have a standardized procedure (website or other) for distributing data. It is therefore often necessary to establish a bilateral research agreement to access the data. In addition, for some organizations, the data is not free and processing fees do apply.

**I think providing more hydrological metrics/signatures would make the database more attractive. Most of the metrics provided are statistical metrics, with somewhat limited use for hydrological (process) studies. There is a wide range of potentially more meaningful signatures (e.g. Addor et al., 2018; or see McMillan, 2020, for a review focusing on process-based signatures).**

We expanded the range of signatures considered, using the TOSHH toolbox recently released, and added the references suggested. The use of this toolbox ensures a consistent calculation of the indices across different datasets and projects. We focus on the indices requiring discharge measurements only, and added signatures about base flow, statistical properties of discharge (skewness, autocorrelation etc.) and also time series of monthly and percentile-based indices.

We choose to focus on the indices requiring discharge only, since there is no consensus on the best precipitation product over Africa, from station-based datasets (CRU, GPCC, REGEN…) or satellite-based products (TRMM, GPM, CHIRPS etc..).

To clearly stress this point in the manuscript we added this section:

To document the mean annual precipitation and evapotranspiration at the catchment scale, the CRU4 dataset has been considered (Harris et al., 2020). However, without long-term and homogeneous ground monitoring networks over the African continent, no best precipitation database could be identified for Africa as a whole (Sylla et al., 2013; Beck et al., 2017; Awange et al., 2019; Satgé et al., 2020). For some regions, such as Northern or Equatorial Africa, there are large differences between different remote sensing or gauged-based precipitation products (Gehne et al., 2016; Harrison et al., 2019; Nogueira, 2020), in particular for extreme precipitation events. This is the reason why we choose to provide only time-averaged precipitation and evapotranspiration, and for a particular application to a given catchment, the user is advised to check the best available precipitation product for that area.

**Similarly, while the database contains some metadata, there is potential to provide much more information. You already calculated catchment shapefiles, which could be shared as well. You could also use the shapefiles to extract more catchment characteristics from global (or African) data products. For example, there are products for P, T, PET, and for many catchment attributes, which would make the database more attractive (in my opinion).**

We re-processed the catchment delineation and we now provide in the dataset the catchment boundaries in a Shapefile format. We also added mean precipitation, temperature and PET from CRU, averaged over a common time period for all basins. We also added the percentage of the main land cover classes from ESA Land cover 2018, as basin descriptors in addition to elevation characteristics.

**Otherwise, it might be a bit hard to compete with recently published datasets that provide time series and various catchment attributes, despite the geographically unique coverage of your database.**

There is no competition here. Different databases offer different kind of data, contributing to open data for scientific research. We hope this new dataset that we are providing will be helpful primary for African students and researchers that often have difficulties in accessing this kind of data.

**Since the original time series are not accessible, there is no way to reproduce the derived indices. But I think that it would be helpful if you could share the code used to create the database.**

We homogenized the computations of the indices using mostly the TOSHH toolbox, that is already available online. We are currently working on a specific code to automate the whole process, with the aim to provide to end-users or basin agencies the capacity to produce the hydrometric indices themselves with new data (mainly by facilitating the pre- and post-processing of data files, since most of the codes are already available in the TOSHH toolbox).

**It would be helpful to discuss a little bit more quantitatively how this database differs for example from GSIM. How many more catchments are in there? This will make it easier for users to see the advantages of that dataset.**

We agree, and added a map and additional text to better explain the data sources and also the comparison with GSIM.

In this ADHI dataset, we combined 672 stations from SIEREM, 794 from the GRDC so a total of 1466 with at least 10 years of data between 1950 and 2018 and a mean record length of 33.3 years and half of the stations have more than 30 years of data.

In GSIM, there are 979 stations in Africa, with record length from 1 year to 110 years until 2015, and a mean record length of 33.8 years.

The major difference is that GSIM will not be updated, when ADHI will.

**There are a few language errors and some unclear sentences. It's mostly minor, but I think proofreading the paper again would be helpful. I made some comments on that in the list below.**

**Minor comments:**

**L:85-86: "carefully checked for quality control". I would rephrase this – you do a quality control, that is, you check for quality. But you don't check for quality control.**

We changed to: "..different sources of data after a quality control".

**L.86-87: This first suggested to me that the database also includes the discharge data, which (unfortunately) it doesn't. I would suggest rephrasing it so that this is clear. The same holds true for the following sentences.**

We changed to: "This new African Database of Hydrometric Indices (ADHI) provides a wide range of hydrometric indices and hydrological signatures"

**L.104: "aquifers" – remove the "s"**

done

**L.126: You mentioned two main reasons, but you then list 3 points.**

Replaced by "several reasons"

**L.134: I would suggest writing 20th century instead of using Roman numerals.**

changed

**L.132-136: That sentence is quite long and could be simplified/split.**

We rephrased the sentence to shorten it.

**L.144: The reference Gnann et al. (2020) doesn't really fit here as the study does not look at climate change or human activities. It would fit better into the next sentence after Westerberg et al. (2020).**

done

**L.181: "A careful inspection" – please be more explicit here.**

We replaced "careful" by "visual"

**L.187: "After this data quality processing step,...". Do you mean all the steps described above, that is the minimum length requirement, checking for duplicate time series, and merging with the GRDC?**

yes

**L.197: Replace "About" with some other word, it sounds a bit awkward.**

We replaced it by 'for'

**L.202-205: I would suggest rephrasing that sentence. I think I understand what you want to say here, but it's a bit unclear.**

We rephrased the sentence to:

For only a few data points in the discharge time series, some obvious errors were detected with daily discharge exceeding by several orders of magnitude the median flow. In these cases, the data has been reported as missing data in an absence of an objective criterion to correct the record.

**L.219-220: "manyfold" instead of "many folds"**

Changed

**L.230 and others: perhaps write southern Africa to clearly distinguish between the south of Africa and South Africa. Technically, the capitalisation should suffice, but it can be a bit confusing.**

Changed, we distinguish between southern Africa, the region, and South Africa, the country.

**L.231 "includes" remove the "s"**

Removed

**L.283: I'm not sure what you mean by "relative indices computed with a base period as reference, such as standardized drought indices." To which indices do you refer to?**

We removed this sentence

**L.309: Here you use "indexes", earlier on "indices"**

We replaced by 'indices'

**L.316: I am not entirely sure about the purpose of this paragraph. It reads like a discussion, which I didn't expect in that section. But it's a bit vague and doesn't really help the reader (in my opinion).**

We moved this part to the conclusion. The main idea of this part was to explain that other composite indices can be computed from the ones supplied in the present ADHI database.

**L.351: "sumary.txt file" is called "summary.tab" in the database**

Changed

**L.360: "contains" – remove the "s"**

removed

**L.380: do you mean "organisations" here rather than "organisms"?**

We replaced by 'institutes'

**L. 418-420: Could you provide links (if they exist) to the GRDC and the SIEREM database websites? (Obviously the data cannot be accessed that way, but providing contact info of the data owners would be helpful.)**

We added the two websites

**Figure 1: In the caption you mention twice that regulated basins are basins with dams.**

We updated figure 1

**References**

Addor, N., Nearing, G., Prieto, C., Newman, A.J., Le Vine, N. and Clark, M.P., 2018.A ranking of hydrological signatures based on their predictability in space. Water Re-sources Research, 54(11), pp.8792-8812.

McMillan, H., 2020. Linking hydrologic signatures to hydrologic processes: A review.Hydrological Processes, 34(6), pp.1393-1409.

These references have been added to justify the selection of indices/signatures.

---

## Author Comment (AC2) · 13 Jan 2021

Dear Dr. Do,

Please find below our responses to your comments.

**This manuscript introduces the African Database of Hydrometric Indices (ADHI), an unprecedented collection of streamflow signatures for Africa. I believe the data product will be greatly appreciated by the regional and global hydrology communities (myself included) as it can potentially fill a significant gap in in-situ records of streamflow and thus can advance hydrologic research over the tropics.**

Thank you for your interest in this dataset.

**Although I fully support the publication of this dataset, I have the feeling that the dataset (and associated manuscript) has been developed in a rush, and thus has missed an opportunity to become a great product that benefits a broader range of users. For example, only time series of annual mean, annual max, and annual 7-d min streamflow indices are published although the authors have done an excellent job in synthesizing and quality-controlling that much daily streamflow data.**

The main goal of submitting this data paper in ESSD is to open the discussion with potential end-users of this dataset, about potential improvements on the short term but also on the long term. Contrary to the Global Streamflow Indices and Metadata Archive (GSIM), this ADHI database will be updated over time, by including recent discharge measurements but also ancillary data to document the catchment characteristics. To provide these hydrometric indices is a first step, the next step would be to provide much more attributes such as those available in the CAMELS dataset (Addor et al. 2017).

According to your recommendation and those of the other reviewer, but also the feedback received from some users who already downloaded the database (240 as in January 2021), we improved the database to better fulfill the end-user needs. The new upgraded version should appear in the online repository in a couple of days.

**I'm recommending a major revision to encourage the authors to improve and make the dataset more attractive to the international community. Below, I listed three major improvements that I strongly suggest the authors consider.**

**1. Expand the streamflow indices that could be accessed publicly.**

**- I appreciate the challenges in data restriction that the authors could have faced, but I think that it is defensible to increase the number of published time-series indices from three (in the current phase) to that described in Gudmundsson et al. (2018) - which I believe the authors have mentioned in their manuscript.**

We agree, following also the recommendations of Reviewer 1, we added more hydrological signatures, using the TOSSH toolbox recently released (https://sebastiangnann.github.io/TOSSH_development/p2_signatures.html), to ensure the homogeneity of the calculation procedures across different datasets.

**- I do think that at the minimum, time series of monthly indices (mean, max, min) would be highly appreciated by the global community to support a wide range of hydro-climatological research.**

We added these time series in the database.

**- I feel that the static percentiles that currently published in the summary text file could be re-processed using the block-window approach (e.g. yearly) to derive time-series that are useful to assess hydrological changes in Africa.**

We added the time series of the percentiles.

**2. Providing catchment shapefiles**

**Figure 3 shows that the authors have also compiled/generated a great collection of catchment boundaries. This is another great asset that could benefit a broad range of end-users. I think publishing this information will not inflict any troubles regarding data policies.**

Indeed, the catchment boundaries were delineated using a public dataset (HydroSheds DEM). As indicated in section 2.4 we were able to compare for some basins the areas calculated with the available metadata. But uncertainties remain both on these metadata from very different organizations, but also on the basin delimitation procedure due to the uncertainty on the coordinates of some stations (as indicated also in the GSIM paper). The automatic relocation procedure of stations applied for the GSIM database is probably not optimal in regions with complex orography or areas of low relief and we experienced the exact same issue.

This is why in the last months we tried to: 1/ collect ancient metadata mostly from field campaigns about catchment maps and stations coordinates (in particular many scanned documents from this portal https://horizon.documentation.ird.fr) 2/ delineate the catchment boundaries within a GIS for all catchments where the automatic delineation did not work. Now the catchment boundaries will be provided in the updated database (in shapefile format).

**3. Although I have not provided any specific comments on the manuscript (as I expect a major revision to make the manuscript become stronger), I have some general comments on the writing that may help better highlight the contribution of this dataset:**

**- The title: please consider some assertion titles such as "The production of seventy-year long streamflow indices for 1500 stations across Africa." This type of title reflects better the usefulness of the AHDI and thus will be more attractive to prospective users.**

This title would be misleading to the readers, since not all the 1500 stations have time series over the last seventy years. We modified the title to mention the time period considered, 1950-2018.

**- Some figures were not associated with an insightful discussion (Figure 1 is not exactly what described in Section 2.1; Figure 5 was completely left out in the discussion). Please expand your discussion regarding any "lesson-learned"**

**working with this dataset. For instance, some discussion about the relationship between the annual precipitation (shown as the background of Figure 1) and annual streamflow (generated by the authors) could be useful; section 4 contains effectively only two lists of bullet points- but could be expanded to include examples of "spurious patterns", substantial local changes, or improvement relative to the GRDC (see below.)**

We re-organized the figures and produced new ones, in particular showing the stations from the different data sources and the link between annual precipitation and mean runoff.

The section 4 is about the content of the database, where we describe the file contents.

Following your recommendation, we expanded the sections 3.1 and 3.2 to include more details (and figures) on the indices computed: the links between runoff and catchment area, mean precipitation, the spatial variability of the different indices in relation to the climatic zone and catchment properties etc. However, we think that a deepened analysis of African hydrology is not in the scope of this data paper, as the suggestion of showing results about local changes (trends). This work would require a deeper analysis of the provided data (and this analysis would not fit in a single paper), in order to study long term trends, water balance components, using different precipitation and evapotranspiration datasets, the relationships with land use and geology - among other possible topics of interest.

**- I also think a map showing improvement of ADHI relative to the GRDC database (perhaps in Section 4) could be useful for end-users. For instance, the authors can classify stations into three categories (i) new stations (relative to GRDC), (ii) extended record stations, and (iii) no improvement. The efforts of the authors to publish this data are greatly commended, and I am very excited about the release of the updated AHDI.**

This is a good idea; the information was already present in the metadata but we included in the revised manuscript a map of GRDC/non-GRDC stations.

It should be noted that we did not merge the station data from SIEREM and GRDC: if the same station exists in the two databases, we kept only the one with the longest records.

---

## Author Response (AR2)

Dear Lukas Gudmundsson, Hong Do,
First, I would like to thank you for the evaluation of our manuscript, and the constructive comments to help improving it. I modified the text to take into account your recommendations. Please find below some more details.
Best regards,
Yves Tramblay

**Dear Yves Tramblay,**
**Many thanks for submitting the revised version of the manuscript presenting the The African Database of Hydrometric Indices (ADHI). One referee has now seen the revised manuscript, who only suggests minor updates. In particular, I suggest considering the suggestion to re-format the lists in section 3.2 and 4 to Tables, since this would significantly increase the readability of your work. Once these and all other referee comments have been addressed, I am happy to consider your contribution for publication in ESSD.**
**Best regards,**
**Lukas Gudmundsson**

Indeed, it make more sense to modify the sections 3.2 and 4 to tables, this has been changed.

**The authors have extended the ADHI substantially in this revision, and I believe this data product (and any future updates) will be highly appreciated by the global community. I don't see any major issues and am recommending this manuscript for publication in ESSD.**

Thanks for the positive feedbacks on the modifications.

**The paper still needs substantial editorial changes, which will perhaps be addressed during the production. However, I have some minor suggestions that the authors might consider in the paper's final form:**
**- Line 90: specify that the stations cover most "climatic regions".**

Added

**- Line 124: any reference/number describing the decline of the Africa river network?**

It is mentioned in Roudier et al. 2014 for West Africa, already cited in the sentence. A reference over all Africa is missing, while this information could be found in several technical reports. This is why we added the Figure 2 showing this decline.

**- Line 141: these two sentences sound fragmented. Perhaps a transitioning sentence would help: "...time series of discharge data. To address these challenges, the focus has been shifted to publishing hydrological indices and signatures, which are useful to...**

We added this transitional sentence

**- Line 152: "a minimum of useful information" - the ADHI deserves more than this humble description.**

We removed "a minimum of"

**- Line 170: please clarify whether the "10 full years" are consecutive.**

We added "not necessarily consecutive"

**- Lines 259-278: it is unclear what the outcome of the catchment quality control procedure was. For instance, how did the authors treat 37 stations with incorrect catchment areas reported in the metadata? Perhaps the authors would want to have some sort of a coding system to indicate catchment quality (if that has been implemented – please report in the manuscript). This information will be welcomed by prospective users.**

Indeed, the sentence "For 37 stations in the SIEREM database, the catchment areas where not correct in the metadata, by comparing the delineated catchments." is misleading and not correct. We did not correct the metadata with computed catchment boundaries but comparing with old technical reports of our institute. So we removed this sentence.

**- Line 323: the distinction between indices and signatures might not be clear to many users – some definitions would help here.**

You are right. We added the following sentence: "While hydrological indices refer to standard statistical metrics, such as the mean, maximum, or percentiles computed from the time series of discharge data, hydrological signatures are metrics describing the hydrological behavior and the dominant processes in a river basin. (Addor et al., 2018). "

**- Section 3: I feel that (i) "Available streamflow signatures and indices derived from daily discharge" might provide more information;**
We modified the section title accordingly.

**(ii) sub-sections 3.1 and 3.2 could be merged into a single sub-heading "Streamflow Signature";**

We did not merge the two sections, since section 3.3 does not describe streamflow signatures but the annual and monthly time series provided.

**(iii) the 25 signatures calculated could be presented in a table form and grouped into different "purposes" if possible.**

Yes, we removed the list and added a table instead. Thanks for this suggestion; the presentation of the indices is indeed better in a table.

**- Section 4: Perhaps this is an editorial decision but the current section reads quite strange (basically two long lists). I am not sure what would be the standard to present the**

**attributes that have been published through ADHI, but reformating these two lists into two tables, each of the attributes could be assigned into a specific category, each table is associated with some description might help this section reads more attractive.**

We agree, we replaced the list with a table.

---

## Author Response (AR3)

Dear Lukas Gudmundsson,

We added in the revised manuscript the description of the columns of the annual and monthly data files. We also added this information in the online repository (this should appear online in a few days).

Thanks for the handling of this manuscript.

Best regards

Yves Tramblay